# *Coxiella burnetii* (Q Fever) in Small Ruminants on Farms in North West Province, South Africa

**DOI:** 10.3390/vetsci12040315

**Published:** 2025-03-31

**Authors:** Katleho N. Mosikidi, Nthabiseng Malekoba Mphuthi, Maruping L. Mangena, David D. Lazarus, Mohammed Sirdar, Nomakorinte Gcebe

**Affiliations:** 1Department of Animal Health, Faculty of Natural and Agricultural Sciences, North West University, Mafikeng 2735, South Africa; knmosikidi14@gmail.com (K.N.M.); nthabiseng.mphuthi@nwu.ac.za (N.M.M.); 2Agricultural Research Council—Bacteriology Laboratory, Onderstepoort Veterinary Research, Private Bag X 05, Onderstepoort, Pretoria 0110, South Africa; gceben@arc.agric.za; 3Agricultural Research Council—Vaccines and Diagnostic Development Programme, Onderstepoort Veterinary Research, Private Bag X 05, Onderstepoort 0110, South Africa; 4Food and Agriculture Organization of the United Nations (Nigeria), Abuja 900001, Nigeria; david.lazarus@fao.org; 5International Society for Infectious Diseases, 867 Boylston Street, Boston, MA 02116, USA; msirdar@isid.org

**Keywords:** Q fever, *Coxiella burnetii*, serology, risk factors, polymerase chain reactions, sequence analysis

## Abstract

Q fever is an agriculturally and economically important disease, it is not considers endemic in South Africa, thus no routine surveillance is conducted resulting in limited or lack of data on the zoonosis. To mitigate limited or lack data on Q fever in South Africa, we conducted a cross-sectional study in small ruminants in farms of North West province of South Africa to determine serological and molecular prevalence of Q fever and associated risk factors. We detected Q fever using both serology and polymerase chain reactions, thus confirming the presence of *C. burnetii* in small ruminants in farms of North West province of South Africa. Our findings will pave way for more in-depth studies on Q fever in other provinces of South Africa and possibly other countries as well. This will create more awareness about the disease, potential vaccine and treatment development which will benefit the scientific community.

## 1. Introduction

*Coxiella burnetii* (*C. burnetii*) of the *Coxieallaceae* family is a Gram-negative bacterium that causes zoonosis, or Q fever [1]. *Coxiella burnetii* infects a broad range of susceptible hosts. These include domestic livestock, wildlife, and non-mammalian species such as reptiles, fish, rats, birds, and ticks, as well as humans [2]. Cattle, sheep, and goats have been associated with the transmission of Q fever to humans through contact with parturient animals [3]. Therefore, there is a risk of infection with *C. burnetii* from the milk products and meat of infected livestock, environmental dispersal in animal excreta, dusts, and during slaughter in abattoirs [4]. This disease presents as mostly asymptomatic in humans with rare clinical manifestations; however, in animals it is associated with reproductive complications such as stillbirth, abortion, repeat breeding, anoestrus, and premature births in infected animals [5,6]. Q fever can be considered an economically important disease as it causes economic losses through reproductive disorders in infected animals [7].

The disease has been reported in animals and humans worldwide, except in New Zealand [8]. In Alabama State, USA, a 7.5% *C. burnetii* DNA prevalence by PCR has been reported in doe goats, indicating widespread distribution of infection in small ruminants [9]. Another study reported 94% and 47% *C. burnetii* seropositivity in goats and sheep, respectively, in Germany. The same study also reported a *C. burnetii* PCR positivity of 8% and 3% in goats and sheep, respectively [10]. The distribution of Q fever and its prevalence in South Africa has been reported recently in cattle in communal farms from Limpopo province, South Africa, with 24.28% and 15.87% serology and molecular prevalence, respectively [11]. A 38% seroprevalence was reported in cattle from the Mnisi region in Mpumalanga province [12], while a seroprevalence of 9.6% in slaughter cattle, sheep, and pigs from Gauteng province of South Africa was observed [13]. Another study in South Africa has also reported the prevalence of the disease in feline and in ticks. In a multi-province study conducted in the Free State, KwaZulu-Natal, and Mpumalanga provinces, a PCR prevalence estimate of 7% of *C. burnetii* was reported in feline hosts, with tick bites being the main mode of transmission [14].

Despite these findings, in South Africa, the disease is not considered to be endemic, as such it is not listed as a controlled disease according to the Animal Disease Act 34 of 1984; therefore, cases of Q fever are not routinely monitored [13]. Sheep and goats are known to be the main reservoirs of Q fever and transmit the disease to other animals through the ingestion of aerosolized particles, direct contact, and tick bites [14]. However, in South Africa, specifically in communal, semi-commercial, and commercial farms in the North West province, there is limited data on Q fever prevalence in sheep and goats. Also, there is limited information on the risk factors associated with Q fever transmission between these animal species as well as in humans. Therefore, this study reports the seroprevalence and molecular prevalence of the disease in sheep and goats from communal (farms organized as one unit and worked by a community under the supervision of the state; agricultural products produced are mainly for subsistence), semi-commercial (part of the agricultural products from farm are for subsistence and the other for market sale), and commercial farms (agricultural products produced on the farms are for market sale rather than subsistence) in the North West province, South Africa, as well as risk factors associated with infection in these animal species.

## 2. Materials and Methods

### 2.1. Ethical Statement

Ethics Approval for this study was granted by the Animal Use Ethics Committee of the Agricultural Research Council-Onderstepoort Veterinary Research and 400 North West University Animal Care, Health and Safety Research Ethics Committee with Ethics Numbers AEC 12–16 and N W U—0 0 5 9 3—1 9—A 5, respectively. All samples were collected with the assistance of State animal health technicians. All the laboratory procedures were conducted under BioSafety Level III (BSL3) conditions at the Agricultural Research Council-Onderstepoort Veterinary Research (ARC-OVR) campus.

### 2.2. Study Area

This is a cross-sectional study conducted in the North West province of South Africa. All four districts of the province, Dr Kenneth Kaunda, Dr Ruth Segomotsi Mompati, Ngaka Modiri Molema, and Bojanala Platinum Districts, were included in the study. Within the districts, the following municipalities were studied: Mahikeng, Greater Taung, Ratlou, Naledi, Kagisano-Molopo, Ramotshere Moiloa, Moretele, and Moses Kotane (Figure 1). The North West province plays an important role in the national agricultural economic activities, and its annual rainfall average varies between 700 mm in the east and less than 300 mm in the west. This province is well known for some of South Africa’s largest cattle herds [https://en.wikipedia.org/wiki/List_of_municipalities_in_the_North_West, 2022, accessed on 5 January 2024].

### 2.3. Sample Size Determination

The sample size was calculated on the assumption of a 50% prevalence of Q fever in the North West province (due to the lack of information on the prevalence of the disease and to account for the maximum possible sample size), with a 5% absolute error and at the 95% level of confidence (95% CI).

The sample size was calculated using the formula by [15]:$$n_{0}=\frac{3.84\times 0.5\times 0.5}{0.0025}=\frac{0.96}{0.0025}=384$$

### 2.4. Sample Collection

In this cross-sectional study, a stratified multistage sampling method was performed, and sampling was conducted in all four seasons of the year; autumn, winter, spring, and summer. The North West province was divided according to districts and then municipalities. Farms or villages (dip-tanks) with or without a previous history of abortion in each municipality were randomly selected. A total number of 421 animals, consisting of 266 goats and 155 sheep, were sampled (Table 1). Three different samples were collected (whole blood, vaginal swabs, and heath scrapings) from sheep and goats. The whole blood samples were collected from the jugular veins using a 20-gauge vacutainer needle and collected in Becton, Dickinson and Company (BD)-Vacutainer^®^ SST^TM^ II Advance 10 mL serum collection tubes (Becton, Dickinson and Company (BD), NJ, USA). Serum was harvested and stored at −20 °C until processing.

Sheath scrapings were collected from bucks and rams using a silicon–rubber pipette that was connected to a syringe. Aspirated material was transferred to the tube containing 4 mL of phosphate-buffered saline solution (PBS) at pH 7.45. The samples were labeled and transported in an insulated cold box to the laboratory and stored at −20 °C until processing.

Vaginal swabs were collected from does and ewes. The collection of the samples was performed using cotton head buds and long swab sticks. The animals were physically restrained by holding their tail out of the way and the cotton head bud was carefully inserted into the vagina. The sample was collected by carefully moving the cotton bud in circular motions, while moving it up and down 5 times. The sample was transferred into a specimen tube labeled using the identification number of the animals and its location. The samples were stored at room temperature (15–30 °C) while they were being transported to the Agricultural Research Council-Onderstepoort Veterinary Research (ARC-OVR) for processing. Samples were stored at −20 °C until further processing. Information about the animals’ breed, age, pregnancy status, and abortion history was obtained from the farmers and farm workers through a verbal questionnaire.

### 2.5. Serological Testing

Evidence of antibodies to *C. burnetii* in serum specimens was investigated using the IDEXX Q fever 2/strip antibody ELISA kits (IDEXX Laboratories, Liebelfld-Bern Switzerland) for the detection of Immunoglobulin G (IgG) antibodies. A total of 421 serum samples from 421 animals were tested using serology according to the manufacturer’s instructions. For the interpretation of the results, a S/P% < 30% represented a negative result, 30% ≤ S/P% < 40% a suspect, while S/P% ≥ 40% represented a positive result [10].

### 2.6. DNA Extraction

A total of 126 samples (122 vaginal swabs and 4 sheath washes) corresponding to seropositive animals were subjected to DNA extraction and PCR amplification of the insertion element of the 146 bp fragment of the transposase gene IS*1111*. For the extraction of DNA from sheath scrapings and vaginal swabs, the High Pure PCR template preparation kit (Roche Diagnostics GmbH, Mannheim, Germany) was used according to the manufacturer’s instructions. A sample volume of 1 mL from sheath scrapings/vaginal swabs in PBS at pH 7.4 was transferred to a 2 mL centrifuge tube and centrifuged at 1467× *g* for 10 min. A volume of the supernatant was discarded, leaving approximately 200 µL to re-suspend the pellet. Briefly, 40 µL of proteinase K and 200 µL of the binding buffer were added to 200 µL of the sample. Following incubation at 70 °C for 10 min, 100 µL of isopropanol was added, and the mixture was then centrifuged at 8000× *g* for 1 min; the flow-through and collection tube were discarded. A total volume of 500 µL of inhibitor removal buffer was added, the mixture was centrifuged at 8000× *g* for 1 min, and the flow-through and collection tube were discarded. This was followed by adding 500 µL of the wash buffer and centrifugation at 8000× *g* for 1 min; the flow-through and collection tube were discarded. Following the addition of 500 µL of wash buffer, the mixture was centrifuged at 8000× *g* for 1 min, and the flow-through was discarded. After centrifuging for 10 s at 13,000× *g*, the collection tube was discarded, and a new tube was added, along with 200 µL of elution buffer, before centrifugation for 1 min at 8000× *g* and storing at −20 °C. The extracted DNA was amplified using *Coxiella* IS*1111* PCR [16].

### 2.7. PCR Detection of C. burnetii

PCR for detection of *C. burnetii* in sheath washes and vaginal swabs was conducted in a 50 µL reaction targeting the multi-copy transposase gene in insertion element, IS*1111* [16], using primers IS*1111*F-5′ CGCAGCACGTCAAACCG3′ and IS*1111*R-5′ TATCTTTAACAGCGCTTGAACGTC3′ [16]. The *Coxiella* gene fragment (gblock) from Integrated DNA Technologies (IDT, Iowa, USA) and water were used as positive and negative controls in the reaction, respectively. The reaction mixture contained 400 nM of each primer, 25 µL of the Ampliqon 2× Taq DNA polymerase Master Mix Red (Ampliqon A/S, Odense, Denmark), and 10 µL of the extracted DNA. PCR amplification was conducted using a BIO RAD T100™ thermal cycler (BIO RAD, Hercules, CA, USA). Cycling conditions consisted of initial denaturation at 95 °C for 15 min, 35 cycles of denaturation at 95 °C for 30 s, annealing at 60 °C for 30 s, and extension at 72 °C for 60 s for 35 cycles. Final extension was carried out at 72 °C for 10 min and amplicons were visualized on a 1.5% *w/v* ethidium bromide-stained agarose gel with an expected size of 146 bp [16], and estimated using Quick-Load^®^ 100 bp DNA Ladder (New England Biolabs, Ipswich, MA, USA).

### 2.8. Sequence Confirmation of C. burnetii

Sequence confirmation of the amplicons was conducted using Sanger sequencing. The IS*1111* PCR products of 20 swabs and sheath washes from 20 animals were sent to Inqaba Biotechnical industries (Pty) Ltd. (Pretoria, South Africa) for sequencing. Sequences were manually edited using the BioEdit Sequence alignment editor (version 7.2.5). Sequences were then analysed on the NCBI BLAST platform for species identification [https://blast.ncbi.nlm.nih.gov/Blast.cgi, accessed on 5 January 2024].

### 2.9. Data Analysis

Continuous variables were reported as frequencies with percentages and categorical variables were reported as percentages with 95% confidence intervals (95% CI). The prevalence was calculated as the proportion of animals that tested positive for antibodies to the Q fever multiplied by 100%, and is presented with a 95% CI. The Chi-square and Fisher’s exact tests were used to determine the association between risk factors and exposure outcomes. A logistic regression model was developed to determine the association of age, species, animal gender, breed, type of farm, the different districts in the North West province, season of the country, pregnancy, and history of abortion with the prevalence of *C. burnetii* antibodies. Data analysis was performed using the OpenEpi—Open—Source Epidemiologic Statistics for Public Health, Version 3.01 www.OpenEpi.com at a 5% level of significance [17]. The agreement between the ELISA and PCR results was assessed using Cohen’s kappa [18]. Data were captured and cleaned in a Microsoft Excel spreadsheet. All analyses, including the correlation between PCR and ELISA tests, were performed using StataCorp. 2015. Stata Statistical Software: Release 14.2 College Station, TX: StataCorp LP. Sequence data were analysed on the NCBI BLAST platform to determine the sequence identity.

## 3. Results

### 3.1. Seroprevalence of Q Fever and Risk Analysis

A total of 421 blood samples were collected from sheep and goats, where 33.96% (143/421) tested positive for specific antibodies against *C. burnetii*, with no significant differences in seropositivity between the two animal species (*p* = 0.094) and among districts (*p* = 0.084) using univariate analysis. The differences in seropositivity were neither significant among breed (*p* = 0.218), sex (*p* = 0.887), age (*p* = 0.372, farm types (*p* = 0.707), seasons of the year (*p* = 0.103), pregnancy status of animals (*p* = 0.264) or abortion history (*p* = 0.347), as demonstrated in Table 1 and Figure 1. No significance in the likelihood of Q fever seropositive among pregnancy status of animals (*p* = 0.264), districts (geographical area), and seasons of the year was observed (Table 1, Figure 2).

### 3.2. Molecular Detection of C. burnetii and Associated Risk Factors

One hundred and twenty-six samples were subjected to PCR detection of *C. burnetii* targeting IS*1111* 146 bp fragment. Out of the 126 samples, 61.11% (77/126) were positive (Table 2, Figure 3). The univariate analysis revealed significant differences in PCR detection of *C. burnetii* between seasons (*p* = 0.001), where most cases were observed in summer and autumn while the least were observed in spring and winter seasons, respectively (Table 2). There were no significant differences in PCR detection of *C. burnetii* between goats and sheep (*p* = 0.148), breeds (*p* = 0.170), sex of animals (*p* = 0.669), age categories (*p* = 1.47), and farm types (*p* = 0.191). Also, no significant difference in C. *burnetii* PCR detection was observed among districts (*p* = 0.097), pregnancy status of animals (*p* = 0.777), or abortion history of the animals (*p* = 0.093), as demonstrated in Table 2.

### 3.3. Sequence Analysis of C. burnetii IS1111

Sequence analysis of 20 *C. burnetii* PCR products showed similarity to *C. burnetii* transposase gene; IS*1111* (Table 3).

### 3.4. Correlation Between ELISA and PCR Results

A higher number of animals (sheep and goats) were positive using PCR, 77/126 (61.11%), compared to ELISA, 60/126 (47.62%). Of the 66 animals that were negative using ELISA, 45 (68.18%) were positive using PCR with a correlation of 31.82% observed (Table 4). A total of 32/60 (53.33%) ELISA-positive samples were also positive using PCR. Out of all the 126 total samples tested by PCR, 28 (57.14%) samples were negative using PCR and positive using an ELISA with an observed 42.86% correlation. Of the overall PCR-positive samples, there was an agreement of 32 (41.56%) positive using an ELISA assay. A comparison of the ELISA and PCR results revealed a fair agreement with Cohen’s kappa (k = 0.22), as shown in Table 4.

## 4. Discussion

In this study, we investigated eight risk factors (species, breed, gender, age category, farm type, season, pregnancy, and history of abortion) associated with Q fever seropositivity in small ruminants in various districts (i.e., Dr Kenneth Kaunda, Dr Ruth Segomotsi Mompati, Ngaka Modiri Molema, and Bojanala Platinum) in the North West Province, South Africa. Although this study is not the first small ruminant (sheep and goat) livestock survey to show serological evidence of antibodies to *C. burnetii* in communal, semi-commercial, and commercial farms in the North West province, South Africa, studies are still limited. Recently, ref. [19] reported a 21.4% Q fever seroprevalence in goats in the Moretele local municipality under Bojanala Platinum district, North West province, South Africa. Another study in Ghana showed an overall Q fever seroprevalence of 21.6% in sheep, cattle, and goats in the Volta region of Ghana [20]. Moreover, ref. [21] reported a 21.4% seroprevalence in dairy goats from commercial farms in the Netherlands. All these observations are lower than the 33.9% seropositivity observed in the present study, suggesting that small ruminants on farms in North West province, South Africa, may also be important reservoirs of *C. burnetii.*

In the current study, no significant differences were observed in seropositivity between sheep and goats (*p* = 0.148) [19]. Other studies reported otherwise. A report by the authors of [22] observed higher Q fever seroprevalence in sheep (1.41%) than in goats (0.76%) in Nandi County, Kenya, with significant differences in seropositivity (*p* = 0.015), suggesting that sheep may be more important reservoirs of *C. burnetii* antibodies than goats. However, other studies have not observed significant differences in seropositivity between sheep and goats. A report by the authors or [23] found insignificant differences in seropositivity between sheep and goats in Nigeria (*p* = 0.734). These findings may suggest that species may not be an important determinant of *C. burnetii* seropositivity, as observed in the study.

We also observed no significant differences in seropositivity according to sex of the animals tested (*p* = 0.669). A similar study in the Moretele district municipality, Bojanala platinum district, North West province of South Africa observed insignificant differences in Q fever seropositivity between female and male goats (*p* = 0.286). These results may mean that the sex of goats may not be an important determinant of seropositivity to *C. burnetii*) [19]. On the other hand, ref. [23] reported significant differences in seropositivity to *C. burnetii* between male and female sheep (*p* = 0.032) and goats in Nigeria (*p* = 0.022) in Southeast Nigeria. A study in northeastern Brazil also reported an association (*p* = 0.0020) between the sex of sheep and *C. burnetii* seropositivity [24]. These results may mean that the sex of sheep and goats is an important determinant of seropositivity to *C. burnetii*. Similarly, in this study no significant difference in seropositivity was observed among breeds. Our findings align with the recent study in Moretele local municipality, Bojanala Platinum district, North West province, South Africa, which did not find any significant difference between breed and seropositivity to *C. burnetii*. This may also suggest that breeds of sheep and goats may not be an important risk factor for seropositivity to *C. burnetii* [19].

In the present study, although not significant, older animals were more likely to be more Q fever seropositive than sheep and goats that were <2.5 years old. This finding is similar to an observation by the authors of [20] who observed that older goats 7–19 months were more likely to have Q fever seropositivity than goats 0–6 months old from Volta region, Ghana. A study by the authors of [25] also showed that animals older than 4 years were more Q fever seropositive than younger animals in Egypt. The high seropositivity in older animals in the current study might be that they have been more exposed to *C. burnetii* in the environment than younger animals. A similar study in goats in Moretele municipality, Bojanala Platinum district of North West Province, South Africa, observed similar results [19].

In addition, by using commercial farms as a base (Table 1), the present study shows that sheep and goats from communal farms are more likely to be *C. burnetii* seropositive than sheep and goats from semi-commercial farms. This result might mean that communal farms are more responsible for *C. burnetii* infections and transmission in small ruminants than semi-commercial and commercial farms. Also, during the summer season sheep and goats from this study were likely to be less seropositive than in both winter and spring seasons, respectively. This observation is in agreement with findings by [26] in dairy cattle in the Hokkaido region, Japan. The higher seroprevalence observed in winter may be because animals gather closer to each other for warmth during cold weather, promoting transmission among the animals. In addition, winter is a dry, windy season in the North West province of South Africa. These conditions promote dispersal of particles, which promote *C. burnetii* infections. *Coxiella burnetii* is transmitted through inhalation of aerosolized particles from infected specimen such as dried shed milk particles, manure, and reproductive organs such as the placenta from infected animals [11,27]. Moreover, we observed in the current study that goats and sheep with no abortion history were more seropositive than those with an abortion history (Table 1). This result was not expected since Q fever causes birth defects including abortions in infected animals. A recent study on cattle in Limpopo province, South Africa, found that cattle with an abortion history were more *C. burnetii* seropositive than those with no history of abortion, contrary to observations in the current study [11].

The present study did not observe significant differences in *C. burnetii* PCR detection between species, breeds of animal, sex, and farm types. A recent study also found no significant differences in *C. burnetii* PCR detection between sex, and farm types in cattle of the Limpopo province farms, South Africa, thus supporting our findings. This suggests that these variables are not important factors in the spread *of C. burnetii* infections [11]. However, the current study observed significant differences in *C. burnetii* PCR detection among districts, seasons of the year, and abortion history of the animals (Table 2). The same study observed that animals with abortion history were more likely to be *Coxiella PCR* positive than ones without abortion history. This result is expected because *C. burnetii* infections are associated with reproductive disorders, such as late abortions and stillbirths, in infected ruminants [11]. We observed a correlation of 0.57 (k = 22) between serology and PCR results in the study, with more animals positive using PCR compared to an ELISA. This correlation is considered fair according to [18]. A similar study on cattle in the Limpopo province farms of South Africa observed different results, where more cattle were more Q fever positive using i-ELISA than PCR [11]. Our observations in the study may mean that more of the sheep and goats were recently infected and shedding *C. burnetii* in their reproductive organs (vagina and penis) as PCR assays have capabilities of detecting recent infections in genomic DNA compared to antibody assays [27]. This is because antibody development in response to *C. burnetii* infections has been reported to occur 2–3 weeks post infection [11]. The presence of PCR-positive and ELISA-negative samples may indicate a current infection and active shedding. This may suggest that IgG antibodies have not been produced yet in response to *C. burnetii* infection or antibodies are below the threshold of detection titers, thus the negative ELSA results [11]. In addition, we observed the presence of ELISA-positive and PCR-negative results. This may suggest chronic *Coxiella* infections in the animals because antibody production in response to *C. burnetii* occurs after 2–3 weeks and antibodies may persist for years [11] without active infection. Also, low pathogen levels in DNA may affect the sensitivity of PCR assays, resulting in negative results [11]. Furthermore, sequence analysis of the *IS1111* PCR products confirmed the detection of *C. burnetii* in all sequenced samples, as shown by the similarity with the transposase gene of *C. burnetii* sequences submitted to the NCBI database. This is consistent with findings in studies in South African livestock and wildlife and cattle farms in Limpopo provinces, respectively, confirming successful PCR detection of the bacterium. This also may mean that the bacterium is widespread among different livestock species and regions of South Africa [1,27]. The present study also demonstrates a fair correlation between ELISA and PCR results. This result is consistent with findings in cattle farms of Limpopo province, South Africa [11].

## 5. Conclusions

The present study successfully detected *C. burnetii* in goats and sheep in farms of the North West province, South Africa. This might not be the first study in the province, but it provided additional information for other regions in the North West. This study also contributed to more information about *C. burnetii* infections in small ruminants in the country, as studies on *C. burnetii* are still limited in South Africa. This information is important in knowing disease status as well as providing awareness about the bacterium in small ruminants. Based on our findings, *C. burnetii* is widespread across goats and sheep in farms in different districts of the North West province, South Africa. There is a need for more in-depth studies on *C. burnetii* in South Africa, including the characterization of the circulating *C. burnetii* genomes, which will provide more data about the disease epidemiology.

## 6. Study Limitations

In this study, a low number of sheath washes and vaginal swabs compared to serum samples were tested due to sample loss.

## Figures and Tables

**Figure 1 vetsci-12-00315-f001:**
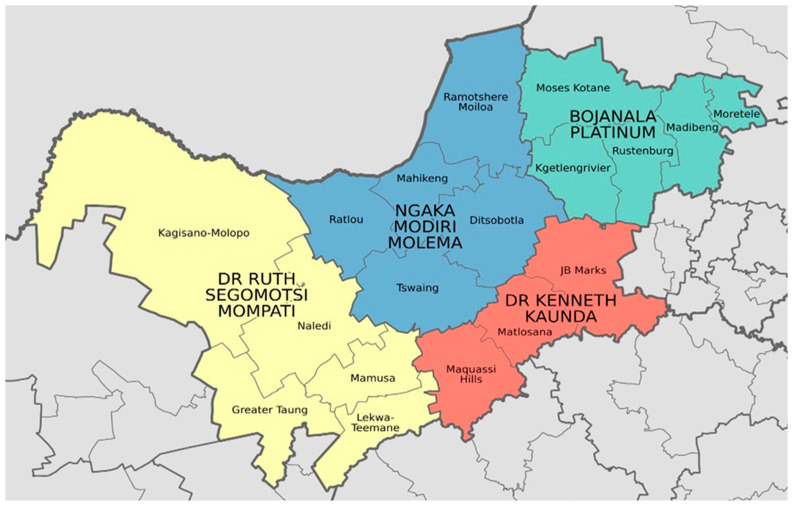
North West province with the 4 districts and the 18 different local Municipalities. [https://en.wikipedia.org/wiki/List_of_municipalities_in_the_North_West, 2022, accessed on 5 January 2024].

**Figure 2 vetsci-12-00315-f002:**
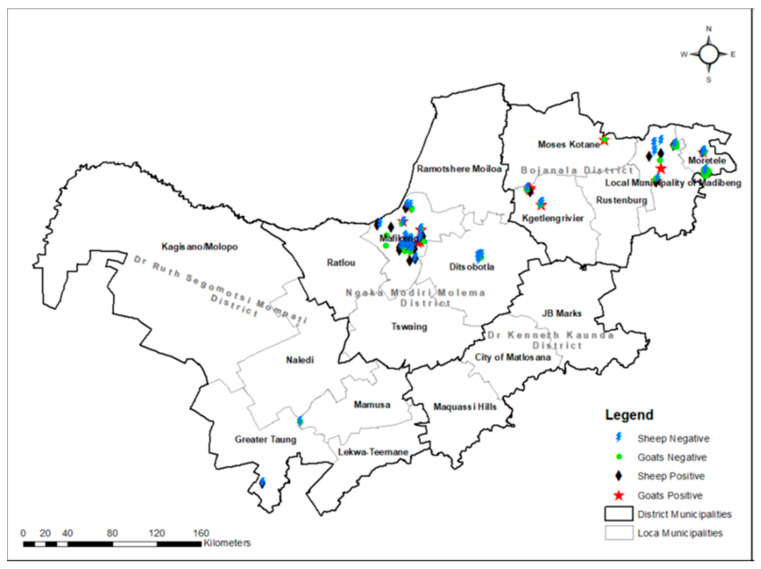
*Coxiella burnetii* antibody-based prevalence by ELISA in various districts of the North West province.

**Figure 3 vetsci-12-00315-f003:**
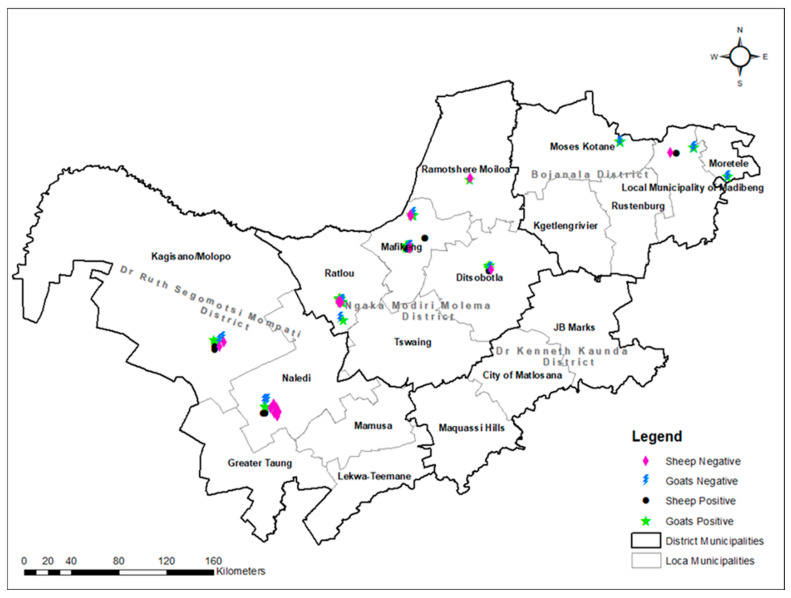
Molecular detection of *C. burnetii* by conventional PCR in various districts and local municipalities of North West province.

**Table 1 vetsci-12-00315-t001:** Animal-level seroprevalence of Q fever and risk factors associated with seropositivity in North West Province of South Africa.

Variables	Total Tested	Number of Positives	Prevalence (%)	Odd Ratio	95% CI	*p*-Value
**Species**						
Goat	266	82	30.82	0.686	0.45–1.04	0.094
Sheep	155	61	39.35	1		
**Breed**						
Boergoat	264	81	30.68	1		
Dorper	105	41	39.05	0.691	0.43–1.11	0.156
White Dorper	44	19	43.18	0.582	0.30–1.11	0.142
Others	8	2	25.00	1.328	0.26–6.72	0.779 ^a^
**Sex**						
Female	395	134	33.92	0.969	0.42–2.23	0.887
Male	26	9	34.62			
**Age category**						
1–2 years	65	27	41.51	1		
2.5–4 years	158	52	32.91	1.448	0.79–2.62	0.286
>4 years	198	64	32.32	1.488	0.83–2.64	0.228
**Farm type**						
Commercial	38	13	34.21	1		
Communal	378	130	34.39	0.992	0.49–2.00	0.875
Semi-Commercial	5	2	40.00	0.780	0.11–5.27	0.800 ^a^
**Season**						
Autumn	136	46	7.44	1		
Spring	197	65	7.25	1.038	0.65–1.65	0.968
Summer	35	18	11.31	0.482	0.23–1.02	0.084
Winter	53	14	5.81	1.424	0.70–2.88	0.420
**Pregnancy status**						
Not Pregnant	400	133	33.25	1.825	0.75–4.41	0.264
Pregnant	21	10	47.62			
**History of abortion**						
Abortion	89	26	29.21	1.319	0.79–2.19	0.347
No abortion	332	117	35.24			

OR = Odd Ratio, CI = Confidence Interval, ^a^ Based on Mid-P exact.

**Table 2 vetsci-12-00315-t002:** Animal level frequency of *C. burnetii* detection using PCR.

Variables	Total Tested	Number Positives	Frequency Occurrence (%)	95% CI	*p*-Value
**Species**					
Goat	68	46	36.51	55.85–77.56	0.148 ^a^
Sheep	58	31	24.60	40.80–65.67	
**Breed**					
Boergoat	68	46	67.64	55.85–77.56	0.170 ^b^
Dorper	46	26	56.52	45.25–69.79	
White Dorper	12	5	41.67	19.33–68.05	
**Sex**					
Female	122	75	61.47	52.62–69.63	0.669 ^a^
Male	4	2	50.00	15.00–85.00	
**Age category**					
1–2 years	3	11	84.62	57.77–96.67	0.147 ^b^
2.5–4 years	68	38	55.88	44.08–67.05	
>4 years	45	28	62.22	47.63–74.89	
**Farm type**					
Communal	6	2	33.33	9.67–70.00	0.191 ^a^
Semi-Commercial	12	75	62.50	53.58–70.65	
**District**					
Bojanala	10	7	70.00	39.68–89.22	0.097 ^b^
Dr Ruth Segomotsi					
Mompati	33	15	45.44	29.84–62.01	
Ngaka Modiri Molema	83	55	66.22	55.58–75.52	
**Season ^i^**					
Autumn	38	28	73.68	57.99–85.03	0.001 ^b^
Spring	53	25	47.16	34.38–60.34	
Summer	16	15	93.75	71.67–98.89	
Winter	19	9	47.36	27.33–68.29	
**Pregnancy status**					
Not Pregnant	124	76	61.29	52.50–69.40	0.777 ^a^
Pregnant	2	1	50.00	9.45–90.55	
**History of abortion**					
**Abortion**	23	10	43.47	25.64–63.19	0.093 ^a^
**No Abortion**	103	67	65.05	55.45–73.56	

^a^ Based on Mid-P exact and ^b^ Based on Chi-Square tests, CI = Confidence Interval ^i^ sampling was performed during the different seasons.

**Table 3 vetsci-12-00315-t003:** Sequence searches against GenBank nucleotide database for closest sequence relative.

Sample ID	BLAST Result: Description of Strain	Accession Number	Percentage Similarity (%)	e-Value	Query Coverage
G-B-Madi	*Coxiella burnetii* strain NKH2 insertion sequence IS*1111*, partial sequence	MF197400.1	79.74	4.00 × 10^−18^	98%
S-B-Gany	*Coxiella burnetii 5*4TI transpose gene, partial cds	MT268532.1	92.57	2.00 × 10^−50^	98%
G-A-Gany	*Coxiella burnetii* isolate Cth 974 transposase gene, partial cds	MK758121.1	91.95	1.00 × 10^−48^	96%
B10-8	*Coxiella burnetii* strain 54T1 transpose gene, partial cds	MT268532.1	86.18	8.00 × 10^−35^	98%
S-C-Stel	*Coxiella burnetii* strain 54T1 transposase gene, partial cd*s*	MT2685321.1	96.62	2.00 × 10^−60^	97%
S-C20-Sebo	*Coxiella burnetii* strain 54T1 transposase gene, partial cds	MT2685321.1	89.47	1.00 × 10^−42^	95%
S-C4-Matl	*Coxiella burnetii* strain 54T1 transposase gene, partial cds	MT2685321.1	89.26	5.00 × 10^−42^	97%
S-4A-Dith	*Coxiella burnetii* strain 54T1 transposase gene, partial cds	MT2685321.1	91.1	6.00 × 10^−46^	98%
S-AD-Lotl	*Coxiella burnetii* clone GaHi12F8 insertion sequence IS*1111*A transposase gene, partial cds	MN094847.1	88.74	6.00 × 10^−41^	98%
S-H1-Kgom	*Coxiella burnetii* insertion sequence IS*1111*A transposase gene, partial cds	KU058956.1	89.04	2.00 × 10^−41^	98%
G-6-Mata	*Coxiella burnetii* clone GaHi12F8 insertion sequence IS*1111*A transposase gene, partial cds	MN094847.1	87.66	1.00 × 10^−38^	98%
S-B20-Mole	*Coxiella burnetii* strain 54T1 transposase gene, partial cds	MT268532.1	92.67	6.00 × 10^−51^	97%
G-A2-Lotl	*Coxiella burnetii* strain 54T1 transposase gene, partial cds	MT268532.1	93.88	3.00 × 10^−53^	96%
S-D1-Lotl	*Coxiella burnetii* Z3055 complete genome	LK937696.1	90.41	1.00 × 10^−42^	96%
G-E-Matl	*Coxiella burnetii* Z3055 complete genome	LK937696.1	89.8	1.00 × 10^−42^	96%
G-X-Masu	*Coxiella burnetii* isolate Cth 974 transposase gene, partial cds	MK758121.1	91.1	6.00 × 10^−46^	96%
G-H4-Kgom	*Coxiella burnetii* isolate Cth 974 transposase gene, partial cds	MK758121.1	88.74	2.00 × 10^−40^	94%
S-AB-Scho	*Coxiella burnetii* isolate INIFAP Cap01 insertion sequence IS110 transposase gene, partial cds	MT462981.1	88.51	3.00 × 10^−38^	98%
G-A-Rooi	*Coxiella burnetii* Cth 974 transposase gene, partial cds	MK758121.1	86.49	2.00 × 10^−35^	95%
G-C-Dith	*Coxiella burnetii* strain 54T1 transposase gene, partial cds	MT2685321.1	91.78	4.00 × 10^−47^	98%

**Table 4 vetsci-12-00315-t004:** Correlation and agreement between serological (ELISA) and molecular detection (PCR) data.

		PCR		Row Marginals	Agreement	Cohen’s Kappa
**ELISA**	Negative	66	49	115	0.57	0.13
	Positive	60	77	137		

## Data Availability

Data are contained within the article.

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
