# Peer review of "Coxiella burnetii (Q Fever) in Small Ruminants on Farms in North West Province, South Africa"

_vetsci, 2025, doi:10.3390/vetsci12040315_

Round 1
Reviewer 1 Report
Comments and Suggestions for Authors
This is an interesting study which reports the prevalence of C. burnetii in North West provinces in South Africa and builds on the data known from that area of the world. It studies a good number of animals, and uses PCR and serology to assess the prevalence of the bacterial pathogen.
It is generally well executed, but suffers from poor English and needs to be proof read by a native speaker prior to resubmission. In some cases it makes it somewhat difficult to follow.
There are some specific points below.
Please ensure that Gram is capitalised as it is someone’s surname
Line 37- causes a zoonosis ….. (reword)
Line 56- maybe in feline hosts? Or in cats?
Line 57-58- ‘tick bites have been reported to contribute an overall PCR prevalence estimate of 7% of C. burnetii in cats and dogs [11].’ Please reword as this doesn’t make sense
Line 85 and 89- is there not a better reference than Wikipedia in here?
Line 150- was this them all or a selection? If it was a selection please say how it was selected
Line 154-168- was this the manufacturer methods? If so you can remove this as you say it was done following manufacturers instructions
Im not sure that you need table 4 as you have it all in the text?
Line 292-293 - suggesting that sheep may be more important reservoirs of C. burnetii antibodies than sheep- this doesn’t make sense so please check it
Line 316- >4 years old were more Q fever seropositive that- more likely to be or higher antibody titres?
Same is true for line 336
Line 337-338 - please reword as this doesn’t make sense
Same is true for line 350, and 353, and 379 and 384.
Comments on the Quality of English LanguageI have made some suggestions above about areas where the manuscript does not make sense. but I strongly suggest sending the report to a native English speaker to check for grammatical issues
Reviewer 2 Report
Comments and Suggestions for Authors
Dear authors,
This is an interesting study on the epidemiology of Q-fever infection in small ruminants in South Africa. The study is well-designed and presented, but some of my comments may help improve the manuscript.
For the Introduction and the Discussion, the authors need to show the dissemination of Q-fever in the international context rather than compare it with previously conducted regional studies. Why did the authors conduct this study if there are so many similar studies within the same area? Also, for the risk factors, the authors should compare them with more studies in this field.
Please indicate how many samples were collected in the M and M?
Were all samples investigated by serology and PCR?
Why were only 126 samples used for DNA extraction?
Please add which factors were considered during the sampling. How had the authors collected animal information, eg. pregnancy or age? Had the authors considered if the animals were delivered recently?
The amount of the provided references is not sufficient.
